# Associations between Multimorbidity and Physical Performance in Older Chinese Adults

**DOI:** 10.3390/ijerph17124546

**Published:** 2020-06-24

**Authors:** Shan-Shan Yao, Xiangfei Meng, Gui-Ying Cao, Zi-Ting Huang, Zi-Shuo Chen, Ling Han, Kaipeng Wang, He-Xuan Su, Yan Luo, Yonghua Hu, Beibei Xu

**Affiliations:** 1Department of Epidemiology and Biostatistics, School of Public Health, Peking University, Beijing 100191, China; yaoshanshan@pku.edu.cn (S.-S.Y.); caoguiying@bjmu.edu.cn (G.-Y.C.); huangzt@pku.edu.cn (Z.-T.H.); Chenzishuo@bjmu.edu.cn (Z.-S.C.); suhexuan@bjmu.edu.cn (H.-X.S.); 1510306218@bjmu.edu.cn (Y.L.); yhhu@bjmu.edu.cn (Y.H.); 2Medical Informatics Center, Peking University, Beijing 100191, China; 3Department of Psychiatry, McGill University, Montreal, QC H3A1A1, Canada; xiangfei.meng@mcgill.ca; 4Department of Medicine, Yale School of Medicine, New Haven, CT 06510, USA; ling.han@yale.edu; 5Graduate School of Social Work, University of Denver, Denver, CO 80208, USA; kaipeng.wang@du.edu

**Keywords:** multimorbidity, grip strength, gait speed, physical performance

## Abstract

*Background*: Evidence on the association between physical performance and multimorbidity is scarce in Asia. This study aimed to identify multimorbidity patterns and their association with physical performance among older Chinese adults. *Methods*: Individuals aged ≥60 years from the China Health and Retirement Longitudinal Study 2011–2015 (*N* = 10,112) were included. Physical performance was measured by maximum grip strength (kg) and average gait speed (m/s) categorized as fast (>0.8 m/s), median (>0.6–0.8 m/s), and slow (≤0.6 m/s). Multimorbidity patterns were explored using exploratory factor analysis. Generalized estimating equation was conducted. *Results*: Four multimorbidity patterns were identified: cardio–metabolic, respiratory, mental–sensory, and visceral–arthritic. An increased number of chronic conditions was associated with decreased normalized grip strength (NGS). Additionally, the highest quartile of factor scores for cardio–metabolic (β = −0.06; 95% Confidence interval (CI) = −0.07, −0.05), respiratory (β = −0.03; 95% CI = −0.05, −0.02), mental–sensory (β = −0.04; 95% CI = −0.05, −0.03), and visceral–arthritic (β = −0.04; 95% CI = −0.05, −0.02) patterns were associated with lower NGS compared with the lowest quartile. Participants with ≥4 chronic conditions were 2.06 times more likely to have a slow gait speed. Furthermore, the odds ratios for the highest quartile of factor scores of four patterns with slow gait speed compared with the lowest quartile ranged from 1.26–2.01. *Conclusion*: Multimorbidity was related to worse physical performance, and multimorbidity patterns were differentially associated with physical performance. A shift of focus from single conditions to the requirements of a complex multimorbid population was needed for research, clinical guidelines, and health-care services. Grip strength and gait speed could be targeted to routinely measure clinical performance among older adults with multimorbidity, especially mental–sensory disorders, in clinical settings.

## 1. Introduction

Multimorbidity is common among older adults, with a prevalence of up to 100% around the world [1] and up to 76.5% among individuals aged ≥60 years in China [2]. It has been demonstrated that multimorbidity is strongly associated with adverse outcomes, such as high mortality [3], increased functional limitation [4], and poor quality of life [5]. However, biomechanical indicators for multimorbidity to measure its severity and potential impact are still not fully understood.

Grip strength and gait speed could provide a valid and reliable measurement of muscle mass, strength and physical performance [6]. Grip strength is a measure of performance of upper limbs and correlates closely with muscle strength, which is considered as one important component of physical fitness and an inexpensive yet strong predictor for disability, morbidity, and mortality in the older population [7,8,9,10]. Additionally, gait speed is a measure that integrates multiple features including balance, strength, and coordination of lower limb function [11]. Poor physical performance was consistently reported to predict negative prognostic trajectories of multimorbidity over time [12,13,14]. On the other hand, western population-based research has also shown that multimorbidity was negatively associated with decreased grip strength [15,16] and gait speed [17,18]. These findings suggest that physical performance, which can be simple and affordable to assess by grip strength and gait speed, serve as an indicator for multimorbidity severity and capture the influence of multimorbidity on physical function among older adults.

Even though China has a large burden of multimorbidity due to its rapidly aging population [19], there is a paucity of research on the association between multimorbidity and physical performance. Studies primarily focus on the relationships between the presence of multimorbidity or the number of chronic conditions and physical performance with little exploration of different patterns of multimorbidity. Patterns of multimorbidity refer to the classification of chronic conditions into different combinations based on associations between the chronic conditions [20]. Previous studies have reported several specific multimorbidity patterns such as the cardiovascular pattern and respiratory pattern that were associated with shared risk factors, medications, and pathological mechanisms [19,21,22,23]. Based on the multimorbidity patterns identified, some studies demonstrated that there were differential associations of multimorbidity patterns with symptom burden, functioning impairment, and decreased health-related quality of life, suggesting that different combinations of chronic conditions may also have variate associations with physical performance [24,25]. Better understanding of how the multimorbidity patterns act as an important risk factor for functional decline can warn older patients to be specifically aware of their functional changes based on their disease combinations. However, we were not aware of any published work examining the associations between multimorbidity and its patterns and physical performance in an Asian population. Therefore, the present study aimed to examine associations of multimorbidity and its patterns with physical performance including grip strength and gait speed, in a nationally representative sample of older Chinese people.

## 2. Materials and Methods

### 2.1. Study Population and Data Collection

This study utilized data from the China Health and Retirement Longitudinal Study (CHARLS) 2011–2015, a national cohort composed of one baseline investigation in 2011 and two follow-up surveys conducted in 2013 and 2015, respectively. The baseline survey of CHARLS used a multistage sampling strategy with 28 provinces, 150 counties/districts, 450 villages/urban communities covering the whole country. The cohort profile of the CHARLS survey have been described elsewhere [26]. The present study included a total of 11,994 participants aged ≥60 (ranged from 60–100) years from three waves of the survey during 2011–2015. After excluding 1182 respondents with missing data for physical performance tests, 10,112 participants were left for final analyses. The ethical review committee at Peking University approved CHARLS (IRB00001052-09076).

### 2.2. Socio-Demographic and Health-Related Variables

Socio-demographic variables included age, sex, marital status, education level (illiterate, can read/write or elementary school, or middle school and above), and residential region (rural or urban). Marital status was classified into married (including cohabited) and others (separated, divorced or widowed). Smoking and drinking habits were determined by the respondents’ self-reported smoking history (never smokers, former smokers, or current smokers) and alcohol consumption history (never drinkers, former drinkers, or current drinkers). Body mass index (BMI) was calculated as weight/height square (kg/m^2^) and was categorized into four groups (<18.5 kg/m^2^, 18.5–23.9 kg/m^2^, 24.0–27.9 kg/m^2^, and ≥28.0 kg/m^2^) [27].

### 2.3. Chronic Conditions and Multimorbidity

There were 18 chronic conditions included in this study. We ascertained 11 self-reported chronic conditions by asking “Have you been diagnosed with the following chronic conditions by a doctor” including dyslipidemia, diabetes, cancer, chronic lung diseases, liver diseases, heart problems, stroke, kidney diseases, digestive diseases, arthritis, asthma, and glaucoma; while hip fracture was reported by participants through asking “Did you have a hip fracture”. Hypertension was defined from either a self-reported physician diagnosis or a blood pressure ≥90/140 mmHg. Participants were considered to have psychiatric conditions if they reported having been diagnosed with emotional, nervous, or psychiatric problems or with depressive symptoms. Depressive symptoms were measured using the Center for Epidemiologic Studies Short Depression Scale (CES-D) with good validity and reliability, and the score of CES-D ranged from 0–30 (Cronbach’s α = 0.81 at baseline) [28]. A CES-D score of ≥10 was used to identify people with depressive symptoms [27]. Impaired cognition was defined as self-reported having been diagnosed with memory-related disease or having poor self-rated memory or having cognitive impairment. Cognitive function was measured by two cognition measures, namely, episodic memory and mental intactness [29]. The total cognition score was the summation of episodic memory and mental intactness scores and ranged from 0–21 (Cronbach’s α = 0.85 at baseline) [30]. A total cognition score of ≤5 (as close as possible to 7% lowest scoring, approximately 1.5 standard deviations below the mean, a generally agreed-upon criterion for relative cognitive impairment) was defined as having cognitive impairment [31]. Vision impairment was defined as self-reported poor eyesight when looking at things both at a distance and up close. Hearing loss was defined as self-reported poor hearing. Definitions of chronic conditions are shown in Appendix A. Missing values of chronic conditions were regarded as “have not been diagnosed with or self-reported not having the chronic condition [32].”

### 2.4. Assessments of Physical Performance

Performance-based measures included grip strength and gait speed. Grip strength was assessed using a hydraulic handgrip dynamometer, Yuejian LW100 dynamometer (Fabrication Enterprises, Nantong Yuejian Body Testing Equipment, Co., Ltd., Nantong, China). Participants were asked to hold the dynamometer with the greatest strength for a few seconds and then released. Each hand was tested twice, alternating hands between trials with a 30 s rest between measurements on the same hand. The grip test was performed in the standing position unless the participant was physically limited and with the elbows being 90 degrees at the side of the body. Participants were excluded if they were unable to hold the handgrip dynamometer and to perform strength testing with both hands. A maximum grip strength (kg) was created from all four attempts, normalized as grip strength (NGS) per body mass (grip strength (kg)/body mass (kg)) [15,33,34].

Participants were asked to walk along a straight 2.5 m walkway twice (back and forth) at their usual pace with aid of a walking stick or cane, elbow crutches, walking frame or others if needed. The examiner used a stopwatch to time the first and the second walk. Gait speed (meter per second) was calculated as the walking distance divided by the time required to complete the task. The average velocity (m/s) out of two tests at one moment was used for data analysis [12]. According to previous studies [35], the gait speed was further categorized as fast gait speed (>0.8 m/s), median gait speed (>0.6–0.8 m/s), and slow gait speed (≤0.6 m/s) to better distinguish the functioning of the participants.

### 2.5. Statistical Analysis

Descriptive analyses of the study population were performed by calculating the frequencies with percentages for discrete variables and means with standard deviation (SD) for continuous variables. To provide nationally-representative estimates, the percentages and sample mean statistics were weighted using sample weights determined by probabilities proportional to size sampling rules, the rate of age eligibility and non-response rate [36].

To determine how chronic conditions tend to group together that exhibited multimorbidity pattern, an exploratory factor analysis was applied. Factor extraction was performed using the principal factor method and a tetrachoric correlation matrix was used to account for the dichotomous nature of the value. We performed the Bartlett’s test of sphericity and the Kaiser–Meyer–Olkin method to investigate sampling adequacy for conducting factor analysis. The number of factors was determined by the interpretability, having an eigenvalue greater than one, as well as the shape of the scree plot. An oblique rotation (oblimin) of factor loading matrices was used to obtain a better interpretability. Factor loadings represented the strength of association between the chronic conditions and latent factors. In this study, we excluded three chronic conditions (cancer, glaucoma and hip fracture) with a prevalence of <3.0% at baseline to achieve better robustness of factor analysis. A composite factor score (standardized to a mean of 0 and SD of 1) for each identified latent factor was then calculated for each participant at each interview, by multiplying respective factor loading derived and each chronic condition (1 for present and 0 for absent). The greater positive values of factor scores referred to a stronger positive association between the chronic condition with each multimorbidity pattern, and greater negative values referred to a stronger negative association. In this study, for ease of interpretation of the results, we used a categorized factor score and the number of chronic conditions for each multimorbidity pattern to describe the status of multimorbidity patterns of the participants. The categorized factor score was calculated by categorizing the standardized factor score of each multimorbidity pattern into quartiles, and the number of chronic conditions in each multimorbidity pattern by summing up all the chronic conditions affecting the participants in the same multimorbidity pattern.

A generalized estimating equation was used to examine the relationships of individual chronic conditions, condition count, and multimorbidity patterns with physical performance including grip strength and gait speed. All variables included in this study were repeatedly measured in 2011/12, 2013 and 2015, and dummy variables were set up for nominal data. Individual chronic conditions were coded as binary variables and summed up to a cumulative number of chronic conditions (0, 1, 2, 3, and ≥4), modeled as multinomial distribution. To assess the associations between multimorbidity patterns and physical performance, the categorized factor score was assumed to follow a multinomial distribution and the number of chronic conditions in each multimorbidity pattern was coded as a continuous variable. Of the two study outcomes, the NGS was modeled as normal distribution and examined as a continuous variable, whereas the gait speed was categorized into three groups of fast gait speed (>0.8 m/s), median gait speed (>0.6–0.8 m/s), and slow gait speed (≤0.6 m/s), and it was modeled as multinomial distribution. Two-sided *p* < 0.05 was considered statistically significant. Analyses were performed using SAS 9.4 (SAS Institute Inc., Cary, NC, USA).

## 3. Results

Table 1 summarizes the baseline socio-demographic, health characteristics, and physical performance of the study population. Overall, the average age of the participants was 67.3 (SD 6.7) years and 50.4% were women. The average number of chronic conditions occurred in the participants was 3.0 (SD 1.9) with the prevalence of participants with two chronic conditions, three chronic conditions, and four chronic conditions or more were 21.9%, 18.8%, and 34.4%, respectively. Participants had an average 28.4 (SD 10.8) kg grip strength, an average 0.5 (SD 0.2) NGS, and an average 0.8 (SD 2.9) m/s gait speed with 39.2% of them having slow gait speed.

The prevalence of chronic conditions were presented in Appendix A. Finally, 15 chronic conditions with a prevalence of ≥3% at baseline were selected to conduct the factor analysis (Table 2). Four multimorbidity patterns were identified: the cardio–metabolic pattern (hypertension, dyslipidemia, diabetes, heart problems, and stroke), the respiratory pattern (chronic lung diseases, and asthma), the mental–sensory pattern (psychiatric conditions, cognition-related conditions, vision impairment, and hearing loss), and the visceral–arthritic pattern (liver diseases, kidney diseases, digestive diseases, and arthritis).

The average grip strength controlling for age, sex and BMI was 30.0, 29.1, 27.8, 27.2, and 26.0 kg, while the average gait speed was 1.00, 0.98, 0.95, 0.95, and 0.93 m/s, for participants with no chronic condition, one chronic condition, two chronic conditions, three chronic conditions, and four chronic conditions or more at their most recent survey, respectively. Associations between the number of chronic conditions and physical performance are shown in Table 3. Compared with participants reporting no chronic condition, those with a single chronic condition and with multiple chronic conditions were associated with a significant decrease in NGS. In addition, compared with participants with a single chronic condition, those with three or more chronic conditions experienced a decreased NGS. For gait speed, participants with multiple chronic conditions had higher odds of having poor gait speed compared with those with no chronic condition and a single chronic condition. In total, the estimates of changes were more pronounced with more chronic conditions.

Table 4 shows the average grip strength and gait speed controlling for age, sex and BMI. The differences for grip strength between Q1 and Q4 of the cardio–metabolic, respiratory, mental–sensory, and visceral–arthritic patterns were 0.9, 1.7, 4.4, and 2.0 kg, while the differences of gait speed were 0.02, 0.03, 0.07, and 0.01 m/s, respectively. Associations between multimorbidity patterns and physical performance are presented in Table 5. Factor scores of four multimorbidity patterns were all statistically significantly associated with decreased NGS. The most pronounced estimate of coefficient was observed in the highest quartile of the factor score for each pattern. The increased number of chronic conditions for each multimorbidity pattern (except the respiratory pattern) were significantly associated with decreased NGS. Compared with participants with factor scores in the lowest quartile, those with factor scores in the higher quartile for the respiratory, mental–sensory, and visceral–arthritic patterns had higher risks of poor gait speed, while significant associations were observed between the increased number of chronic conditions in the four patterns and higher odds of slow gait speed (*p* < 0.05).

## 4. Discussion

The present study identified the differential associations of four distinguished multimorbidity patterns with physical performance in a nationally representative cohort of Chinese aged 60 years and older. In general, weaker grip strength and slower gait speed were more likely in multimorbid participants compared with people without chronic conditions or with a single chronic condition. Four multimorbidity patterns were identified (cardio–metabolic, respiratory, mental–sensory, and visceral–arthritic) and consistently associated with decreases in grip strength and a higher risk of poor gait speed, with the most pronounced association for the mental–sensory pattern.

Multimorbidity was prevalent among older Chinese people. In this study, we observed similar multimorbidity patterns to those reported previously, and common antecedents and disease pathways may explain these patterns [19,21,22,23]. A large body of literature has reported the cardio–metabolic pattern and its potential explanations [21,37]. Regarding the respiratory pattern, as reported previously, asthma is closely related to the increase in risk of chronic lung disease [38,39]. For the mental–sensory pattern, a small amount of evidence supported a positive correlation between sensory impairment and mental disorders [40,41]. In addition, consistent with previous findings, we found a strong association between kidney diseases, liver diseases, digestive diseases and arthritis, which made up the visceral–arthritic pattern [42,43,44,45].

There was an inverse association between the number of chronic conditions and NGS in this study, which is in line with previous studies of older populations from the US, Brazil, Ghana, India, Mexico, Russia, and South Africa using unadjusted grip strength [10,14,15,16]. Moreover, we found that compared with older people without chronic conditions, the average gait speed for multimorbid people decreased by 0.05–0.07 m/s, which exceeded the best estimate of small meaningful change for gait speed found previously [46]. In addition, older adults with multimorbidity had a higher risk of slower gait speed in this study. One previous study has also reported slower speed gaits observed across the spectrum of multimorbidity in older Peruvian people without functional dependency [18]. Multiple chronic conditions and poor physical performance, including weaker grip strength and slower gait speed, may be linked through potential pathways such as nutritional depletion, systemic inflammation, chronic oxidative stress, and physical inactivity [10,47,48]. Frailty, an age-related clinical syndrome of decreased resilience to internal and external stressors, may partially mediate the association between multimorbidity and physical performance [49]. In addition, it was corroborated that multimorbidity was a major determinant of frailty syndrome [50], while physical performance was often considered to be an important component of frailty [51]. Furthermore, risks of grip strength decline and poor gait speed for participants with multiple chronic conditions were also higher compared to those with a single chronic condition. This suggests that there may be synergistic interactions between chronic diseases, which can trigger and increased risk of functional decline [33]. Given the significant association between physical function and disability, performance-based measures, including grip strength and gait speed, could be used as warning signals in the disablement process of aging and a pre-clinical indicator of disability among older populations with multimorbidity [9,18,52,53]. Furthermore, muscle-strengthening activities may be an alternative solution to prevent and intervene the occurrence of multimorbidity [13,31,54].

Notably, we observed some differences in associations between physical performance and various multimorbidity patterns. The cardio–metabolic pattern had the most pronounced association with NGS, though only a small change of grip strength was observed in this pattern, which suggests that grip strength modeled relative to body mass can be used as a biomarker for obesity-related cardiometabolic chronic conditions [34]. Besides, in line with recent findings from Denmark [20], we also observed greater changes in the mental–sensory pattern than in other patterns in terms of grip strength and gait speed, as well as a robust adjusted association between this pattern and the two measures. This could be explained by the fact that cognitive impairment and vision impairment in the mental–sensory pattern were closely associated with a decline in mobility [20]. In addition, depressive symptoms play an important role in disability, while hypertension and arthritis do not, which might be partially attributable to the difference in treatment and surveillance [55]. Depression in the mental–sensory pattern may be more difficult to be identified among older adults, especially in conjunction with age-related cognitive and sensory changes that are generally seen as a normal phenomenon in older adults [55]. Consequently, the identification and treatment of mental–sensory disorders are likely to be deferred, leading to negative consequences on physical performance. The findings provide evidence that grip strength and gait speed could be used as not only convenient, noninvasive and valid measurements in the general practice for prevention or recovery efforts, but also as prognostic indicators for sensory impairment and mental disorders in treatment [16,31,56].

The present study has a number of strengths. This is one of the first studies to report multimorbidity and physical performance in a national representatively community-based sample of older Chinese people. This study also included the majority of chronic conditions with long-term impacts on health status among the older population [21]. The four multimorbidity patterns were derived by factor analysis, which allows chronic conditions to cross-load and facilitates a better understanding of how chronic conditions naturally group together, without pre-conceived assumptions [24]. Our study measured physical functioning based on performance-based indicators, which are objective, reliable and can better discriminate physical functioning, especially among community dwelling well-functioning older adults [9]. Nevertheless, our study has several shortcomings. First, the majority of chronic conditions included in this study were self-reported, which may be subject to recall bias and information bias. Second, we were limited to the data available from the survey and, as such, there was no information about disease severity, history of clinical services and care, or other muscular-skeleton conditions closely related to physical performance. Our findings cannot rule out the potential influences of the above-mentioned factors in the associations. Third, although longitudinal data were used in this study, data on age onset of these chronic conditions were not available, which limited causal inference between the presence of multimorbidity and physical performances. Last, participants who did not complete the physical performance assessments and were excluded in this study and they were more likely to be older (mean age 68.7 years vs. 66.9 years), cognitive-impaired (prevalence 60.1% vs. 52.8%) and less likely to be multimorbid (prevalence 71.0% vs. 74.9%) than those included, which might limit the generalizability of our findings.

## 5. Conclusions

The increased number of chronic conditions negatively influenced physical performance, and multimorbidity patterns were differentially associated with physical performance. The four multimorbidity patterns identified were all associated with worse physical performance. The association was more pronounced and stable for mental–sensory disorders. Our study suggests an urgent need for health-care services to better meet the needs of older populations by considering the complexity of their multimorbidity. Our findings also have important clinical and practical implications for monitoring the severity of multimorbidity in older people using convenient and safe measurement methods, as well as for developing interventions to prevent adverse health outcomes associated with multimorbidity. Future research should pay close attention to the underlying mechanisms of the relationship between physical performance and multimorbidity, especially multimorbidity in different domains. From public health perspectives, valid and easy-to-use measurements of physical performance are warranted as alternative indicators of multimorbidity, and validated cut-off values for the measurements might be of great help in clinical settings.

## Figures and Tables

**Table 1 ijerph-17-04546-t001:** Baseline information of characteristics of participants (N = 10,112).

Characteristics	*n* (%)
Age (years)	
60–69	7147 (68.0)
70–79	2385 (25.0)
≥80	580 (7.0)
Sex, Women	5089 (50.4)
Body mass index (kg/m^2^)	
<18.5	952 (9.3)
18.5–23.9	5274 (50.6)
24.0–27.9	2818 (28.9)
≥28.0	1068 (11.2)
Marital status	
Married	8174 (78.4)
Others	1938 (21.6)
Education levels	
Illiterate	3414 (34.2)
Can read/write or elementary school	4222 (43.0)
Middle school and above	1952 (22.8)
Residential regions	
Rural	6125 (52.8)
Urban	3987 (47.2)
Smoking status	
Nonsmokers	5759 (57.7)
Former smokers	1524 (15.5)
Current smokers	2829 (26.8)
Alcohol consumption	
Nondrinkers	5287 (52.7)
Former drinkers	3745 (37.4)
Current drinkers	1071 (9.9)
Number of chronic conditions (n)	
0	704 (7.0)
1	1781 (17.9)
2	2163 (21.9)
3	1915 (18.8)
≥4	3549 (34.4)
Gait speed (m/s)	
Fast	2732 (27.2)
Median	3499 (33.6)
Slow	3881 (39.2)

Note: Values are presented as No. (percentages) unless otherwise indicated. No. was unweighted, while percentages were weighted population estimates which incorporated the survey weight variable that was constructed according to sampling probabilities, the rate of age eligibility and non-response.

**Table 2 ijerph-17-04546-t002:** Factor loadings of chronic conditions for the four multimorbidity patterns.

Chronic Conditions	Factor ^a^			
	Cardio–Metabolic Pattern	Respiratory Pattern	Mental–Sensory Pattern	Visceral–Arthritic Pattern
Hypertension	**0.67**	0.02	0.10	−0.21
Dyslipidemia	**0.72**	0.11	−0.12	0.35
Diabetes	**0.74**	0.14	−0.08	0.15
Heart problems	**0.53**	0.41	0.00	0.42
Stroke	**0.59**	0.13	0.31	0.02
Chronic lung diseases	−0.06	**0.90**	0.09	0.11
Asthma	0.04	**0.91**	0.07	−0.06
Psychiatric conditions	0.05	0.26	**0.54**	0.35
Cognition-related conditions	−0.04	0.11	**0.78**	0.02
Vision impairment	0.09	0.25	**0.70**	0.15
Hearing loss	0.06	0.20	**0.70**	0.13
Kidney diseases	0.29	0.37	0.08	**0.53**
Liver diseases	0.12	0.12	0.03	**0.70**
Digestive diseases	−0.11	0.31	0.20	**0.70**
Arthritis	−0.01	0.29	0.31	**0.55**

Note: Kaiser–Meyer–Olkin value = 0.67; Bartlett’s test of sphericity *p* < 0.001. ^a^—Factor loadings indicate the strength of association between each variable and each factor, with a factor loading of ≥0.5 indicated in bold.

**Table 3 ijerph-17-04546-t003:** Associations between multimorbidity and physical performance among older Chinese people (*N* = 10,112).

No. of Chronic Conditions	NGS ^a^	Categorized Gait Speed ^b^
**Compared with no Chronic Condition**	**Adjusted β Coefficient (95% Confidence Interval)**	**Adjusted Odds Ratio (95% Confidence Interval)**
0	Reference	Reference
1	**−0.01 (−0.03–−0.003)**	1.23 (0.97–1.55)
2	**−0.04 (−0.05–−0.03)**	**1.50 (1.19–1.88)**
3	**−0.04 (−0.05–−0.03**)	**1.73 (1.37–2.19)**
≥4	**−0.06 (−0.07–−0.05)**	**2.06 (1.65–2.57)**
Compared with single chronic condition		
1	Reference	Reference
2	−0.01 (−0.02–0.001)	**1.22 (1.03–1.43)**
3	**−0.02 (−0.03–−0.003)**	**1.41 (1.19–1.67)**
≥4	**−0.04 (−0.06–−0.03)**	**1.67 (1.44–1.95)**

Note: ^a^—Models were adjusted for age, sex, marital status, education, residential region, smoking status, and alcohol consumption with normalized grip strength (NGS) as a continuous variable. ^b^—Models were adjusted for age, sex, marital status, education, residential region, smoking status, alcohol consumption, and body mass index with categorized gait speed considered as a multinomial variable. Boldface indicates statistical significance (*p* < 0.05).

**Table 4 ijerph-17-04546-t004:** Predicted physical performance for multimorbidity patterns among older Chinese adult participants of the most recent survey (N = 10,112).

Quartiles of Factor Scores	Multimorbidity Patterns			
	Cardio–Metabolic	Respiratory	Mental–Sensory	Visceral–Arthritic
	Grip strength (kg) ^a^			
Q1	27.8	28.5	30.0	28.6
Q2	27.8	27.9	27.5	27.3
Q3	27.3	26.5	26.5	27.3
Q4	26.9	26.8	25.6	26.6
	Gait speed (m/s) ^a^			
Q1	0.96	0.97	0.99	0.96
Q2	0.96	0.96	0.95	0.94
Q3	0.94	0.94	0.94	0.95
Q4	0.94	0.94	0.92	0.95

Note: ^a^ —Adjusted for age, sex, and body mass index.

**Table 5 ijerph-17-04546-t005:** Associations between multimorbidity patterns with physical performance among older Chinese adults (N = 10,112).

Measures of Multimorbidity Pattern	Multimorbidity Patterns			
	Cardio–Metabolic	Respiratory	Mental–Sensory	Visceral–Arthritic
Quartiles of factor scores	NGS ^a^ Adjusted β coefficient (95% confidence interval)			
Q1	Reference			
Q2	−0.01 (−0.01–0.003)	**−0.02 (−0.04–−0.01)**	**−0.01 (−0.02–−0.01)**	**−0.02 (−0.04–−0.01)**
Q3	**−0.03 (−0.03–−0.02)**	**−0.02 (−0.03–−0.01)**	**−0.03 (−0.04–−0.02)**	**−0.02 (−0.03–−0.004)**
Q4	**−0.06 (−0.07–−0.05)**	**−0.03 (−0.05–−0.02)**	**−0.04 (−0.05–−0.03)**	**−0.04 (−0.05–−0.02)**
No. of chronic conditions	**−0.03 (−0.03–−0.02)**	−0.003 (−0.01–0.01)	**−0.01 (−0.02–−0.01)**	**−0.01 (−0.01–−0.003)**
Quartiles of factor scores	Categorized gait speed ^b^ Adjusted odds ratio (95% confidence interval)			
Q1	Reference			
Q2	1.02 (0.88–1.17)	**1.19 (1.03–1.38)**	**1.25 (1.07–1.46)**	**1.24 (1.06–1.45)**
Q3	1.01 (0.87–1.18)	**1.43 (1.24–1.65)**	**1.52 (1.31–1.76)**	**1.36 (1.17–1.57)**
Q4	**1.26 (1.07–1.49)**	**1.72 (1.48–1.99)**	**2.01 (1.72–2.33)**	**1.53 (1.31–1.78)**
No. of chronic conditions	**1.09 (1.02–1.17)**	**1.20 (1.08–1.34)**	**1.29 (1.23–1.36)**	**1.09 (1.02–1.16)**

Note: ^a^—Models were adjusted for age, sex, marital status, education level, residential region, smoking status, alcohol consumption with normalized grip strength (NGS) as a continuous variable. ^b^—All models were adjusted for age, sex, marital status, education level, residential region, smoking status, alcohol consumption, and body mass index with categorized gait speed used as a multinomial variable. Number of chronic conditions was used as a continuous variable and boldface indicates statistical significance (*p* < 0.05).

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
