# Peer review of "Associations between Multimorbidity and Physical Performance in Older Chinese Adults"

_ijerph, 2020, doi:10.3390/ijerph17124546_

Round 1

Reviewer 1 Report

Comments to the Authors:

The current study investigates the associations between mulitimorbidity and physical performance in older Chinese adults (>60 years old) from the China Health and Retirement Longitudinal Study.  The authors evaluated the relationship of grip strength and gait speed to the number of self-reported physical conditions in over 10,000 patients.  Based on the reported, the authors categorized multimorbidity into 3 patterns:  cardiometabolic, respiratory, mental-sensory, and visceral-arthritic.  In summary, the increased number of multimorbidity patterns was associated with lower gait speed and reduced grip strength. 

Comments, Concerns, and Suggestions:

  1. Although the data set is quite impressive, the major finding of decreased physical performance with an increased number of physical maladies is not surprising and rather expected. The novelty of the study is limited to the Asian population.  Can the authors expand on the importance and significance of the present study and differentiate the findings further from what is already known?
    1. Perhaps the authors can make some comparisons of these findings to other populations?

  1. The rationale for the gait speed categories is referenced to a published paper in 75-year-old patients. This patient population is younger (~67 yo).  Can the authors explain the rationale for using these categories?  In fact, the gait speeds reported at much lower than typical 60-70 yo (~1.24-1.34 m/s).
    1. The “visceral-arthritic pattern” only accounts for arthritis. Given that the gait speed is much lower, did the authors gather any additional information on other musculo-skeletal conditions?  Perhaps, any evidence of sarcopenia?
    2. Predictive equations for gait speed using height. Was this variable consider in the analysis of gait speed?  The authors might consider providing further discussion on this point.
  2. The sample population contains ~50% females. However, did the authors further analyze any potential sex differences in the population.  Are females any more prone to physical limitations than males?  This might help differentiate this study and increase the potential significance.
  3. The authors might consider providing any data displaying the prevalence of each of the individual conditions. For example, out of the ~10,000 patients, how many report hypertension, diabetes, stroke, etc.

Author Response

Thanks for your valuable comments. We have provided point-to-point responses to the the comments in this letter and made the corresponding changes, all highlighted in red in the text, to the revised manuscript.

Comments to the Authors:

The current study investigates the associations between multimorbidity and physical performance in older Chinese adults (>60 years old) from the China Health and Retirement Longitudinal Study. The authors evaluated the relationship of grip strength and gait speed to the number of self-reported physical conditions in over 10,000 patients.  Based on the reported, the authors categorized multimorbidity into 3 patterns:  cardiometabolic, respiratory, mental-sensory, and visceral-arthritic. In summary, the increased number of multimorbidity patterns was associated with lower gait speed and reduced grip strength.

Comments, Concerns, and Suggestions:

Although the data set is quite impressive, the major finding of decreased physical performance with an increased number of physical maladies is not surprising and rather expected. The novelty of the study is limited to the Asian population. Can the authors expand on the importance and significance of the present study and differentiate the findings further from what is already known?

Response: Thanks for the valuable comment. Though the association between an increased number of chronic conditions and physical maladies was investigated previously, this study added new knowledge on the associations between multimorbidity patterns and physical performance. Previous studies have reported that there were differences in the multimorbidity patterns between countries with different disease spectrum and socio-economic environment [1] and that clinical guidelines targeted to specific population was necessary. Therefore, studies on the association of multimorbidity and multimorbidity patterns with physical performance based on a specific population with distinct multimorbidity status have implications for clinical settings and public health management.

Perhaps the authors can make some comparisons of these findings to other populations?

Response: Thanks for your suggestion. We have made some comparisons of the findings to other populations. (Page 10 Paragraph 3-4)

The rationale for the gait speed categories is referenced to a published paper in 75-year-old patients. This patient population is younger (~67 yo). Can the authors explain the rationale for using these categories? In fact, the gait speeds reported at much lower than typical 60-70 yo (~1.24-1.34 m/s).

Response: Thanks for the insightful comment. It is true that the gait speed categories was referenced to the methodology of one study conducted in 75-year-old individuals, which might be problematic. After reviewing more studies, we found that there were no official criteria for gait speed classification. Some other studies categorized gait speed into two groups by the cut-off value of 0.8 m/s [2-5]; or into three groups by 0.8 m/s and 1.0 m/s[6]; or into four groups by 1.0 m/s, 0.8m/s, and 0.6 m/s[7]. Therefore, to make the findings comparable to previous studies and considering of the mean gait speed of 0.77 m/s as well as median gait speed of 0.66 m/s, we reconsidered 0.8 m/s and 0.6 m/s, approximate tertiles, as the gait speed categories to obtain an equivalent sample size of subgroups. Besides, though the average gait speed of our study population was much lower than 1.24-1.34 m/s, it was also similar to some other studies[8-13]. Maybe it can be owing to the difference of social environment between countries and habits between populations.

The “visceral-arthritic pattern” only accounts for arthritis. Given that the gait speed is much lower, did the authors gather any additional information on other musculo-skeletal conditions? Perhaps, any evidence of sarcopenia?

Response: Thanks for the insightful comments. Actually, we were not able to include more information about other muscular-skeletal conditions in this study due to the data availability, which was one of the limitation of this study and we have added it into the Discussion. (Page 11 Line 286-287)

Predictive equations for gait speed using height. Was this variable consider in the analysis of gait speed? The authors might consider providing further discussion on this point.

Response: Thanks for the insightful suggestion. We have used BMI in the analysis, which comprises information about height and weight, and it was reported that BMI have significant association with gait speed [14, 15].

The sample population contains ~50% females. However, did the authors further analyze any potential sex differences in the population? Are females any more prone to physical limitations than males? This might help differentiate this study and increase the potential significance.

Response: Thanks for the valuable suggestion. We have done further analysis on the associations of multimorbidity and its patterns with physical performance stratified by sex. The results were presented in the supplementary material. We find little sex heterogeneity between females and males, though some difference in the p-value of statistical significance. The potential reason might be that the study population was relatively young with better health status, and the follow-up period was not long enough. Therefore, the sex difference was hard to detect.

The authors might consider providing any data displaying the prevalence of each of the individual conditions. For example, out of the ~10,000 patients, how many report hypertension, diabetes, stroke, etc.

Response: Thanks for the insightful suggestion. We have presented the prevalence of individual conditions in the supplemental material.

  1. Violan, C., Q. Foguet-Boreu, G. Flores-Mateo, C. Salisbury, J. Blom, M. Freitag, L. Glynn, C. Muth, and J. M. Valderas. "Prevalence, Determinants and Patterns of Multimorbidity in Primary Care: A Systematic Review of Observational Studies." Plos One 9, no. 7 (2014).
  2. Odden, M. C., A. E. Moran, P. G. Coxson, C. A. Peralta, L. Goldman, and K. Bibbins-Domingo. "Gait Speed as a Guide for Blood Pressure Targets in Older Adults: A Modeling Study." J Am Geriatr Soc 64, no. 5 (2016): 1015-23.
  3. Marengoni, A., S. Bandinelli, E. Maietti, J. Guralnik, G. Zuliani, L. Ferrucci, and S. Volpato. "Combining Gait Speed and Recall Memory to Predict Survival in Late Life: Population-Based Study." J Am Geriatr Soc 65, no. 3 (2017): 614-18.
  4. Vetrano, D. L., D. Rizzuto, A. Calderon-Larranaga, G. Onder, A. K. Welmer, C. Qiu, R. Bernabei, A. Marengoni, and L. Fratiglioni. "Walking Speed Drives the Prognosis of Older Adults with Cardiovascular and Neuropsychiatric Multimorbidity." American Journal of Medicine 132, no. 10 (2019): 1207-15 e6.
  5. Bahat, G., C. Kilic, B. Ilhan, M. A. Karan, and A. Cruz-Jentoft. "Association of Different Bioimpedanciometry Estimations of Muscle Mass with Functional Measures." Geriatr Gerontol Int 19, no. 7 (2019): 593-97.
  6. Ensrud, K. E., L. Y. Lui, L. Langsetmo, T. N. Vo, B. C. Taylor, P. M. Cawthon, M. L. Kilgore, C. E. McCulloch, J. A. Cauley, M. L. Stefanick, K. Yaffe, E. S. Orwoll, J. T. Schousboe, and Group Osteoporotic Fractures in Men Study. "Effects of Mobility and Multimorbidity on Inpatient and Postacute Health Care Utilization." J Gerontol A Biol Sci Med Sci 73, no. 10 (2018): 1343-49.
  7. Demakakos, P., R. Cooper, M. Hamer, C. de Oliveira, R. Hardy, and E. Breeze. "The Bidirectional Association between Depressive Symptoms and Gait Speed: Evidence from the English Longitudinal Study of Ageing (Elsa)." Plos One 8, no. 7 (2013): e68632.
  8. Busch Tde, A., Y. A. Duarte, D. Pires Nunes, M. L. Lebrao, M. Satya Naslavsky, A. dos Santos Rodrigues, and E. Amaro, Jr. "Factors Associated with Lower Gait Speed among the Elderly Living in a Developing Country: A Cross-Sectional Population-Based Study." BMC Geriatr 15 (2015): 35.
  9. Imms, F. J., and O. G. Edholm. "Studies of Gait and Mobility in the Elderly." Age Ageing 10, no. 3 (1981): 147-56.
  10. Studenski, S., S. Perera, D. Wallace, J. M. Chandler, P. W. Duncan, E. Rooney, M. Fox, and J. M. Guralnik. "Physical Performance Measures in the Clinical Setting." J Am Geriatr Soc 51, no. 3 (2003): 314-22.
  11. Stringhini, S., C. Carmeli, M. Jokela, M. Avendano, C. McCrory, A. d'Errico, M. Bochud, H. Barros, G. Costa, M. Chadeau-Hyam, C. Delpierre, M. Gandini, S. Fraga, M. Goldberg, G. G. Giles, C. Lassale, R. A. Kenny, M. Kelly-Irving, F. Paccaud, R. Layte, P. Muennig, M. G. Marmot, A. I. Ribeiro, G. Severi, A. Steptoe, M. J. Shipley, M. Zins, J. P. Mackenbach, P. Vineis, M. Kivimaki, and Lifepath Consortium. "Socioeconomic Status, Non-Communicable Disease Risk Factors, and Walking Speed in Older Adults: Multi-Cohort Population Based Study." BMJ 360 (2018): k1046.
  12. Newman, A. B., J. L. Sanders, J. R. Kizer, R. M. Boudreau, M. C. Odden, A. Zeki Al Hazzouri, and A. M. Arnold. "Trajectories of Function and Biomarkers with Age: The Chs All Stars Study." Int J Epidemiol 45, no. 4 (2016): 1135-45.
  13. Wei, M. Y., M. U. Kabeto, K. M. Langa, and K. J. Mukamal. "Multimorbidity and Physical and Cognitive Function: Performance of a New Multimorbidity-Weighted Index." J Gerontol A Biol Sci Med Sci 73, no. 2 (2018): 225-32.
  14. Tabue-Teguo, M., K. Peres, N. Simo, M. Le Goff, M. U. Perez Zepeda, C. Feart, J. F. Dartigues, H. Amieva, and M. Cesari. "Gait Speed and Body Mass Index: Results from the Ami Study." Plos One 15, no. 3 (2020): e0229979.
  15. Windham, B. G., M. E. Griswold, W. Wang, A. Kucharska-Newton, E. W. Demerath, K. P. Gabriel, L. A. Pompeii, K. Butler, L. Wagenknecht, S. Kritchevsky, and T. H. Mosley, Jr. "The Importance of Mid-to-Late-Life Body Mass Index Trajectories on Late-Life Gait Speed." J Gerontol A Biol Sci Med Sci 72, no. 8 (2017): 1130-36.

Reviewer 2 Report

Deat authors!
I have read the manuscript titled “The association between multimorbidity and physical performance in older Chinese adults.” The document aims to examine the association of multimorbidity and its patterns with physical performance measured by grip strength and gait speed in a nationally representative sample of older Chinese.

The manuscript theme is novel and relevant. The introduction is clear and makes a case for the study. The aim is identifiable and well described. The method section is neat and explains all the relevant variables for the study. However, the discussion underestimates the limitations of the research and lacks organization. Find specific comments below:

Abstract:

Please include headings for each section fo the abstract for readability. Similarly, matching the aim in the abstract with the objective in the manuscript. Therefore, re-organizing the results according to the objective of the study (patters identified and their association with gait speed and grip strength).

Introduction:

Page 1, paragraph 1, sentence 2. Please verify the word « inverse ».

Method:

Specify that the design of the study reported is cross-sectional, including a subsample of participants of a follow-up survey.

It would be beneficial if the authors indicate which measurement data were included in the study (2011, 2013 or 2015).

The supplementary table does not add to the information included in the “Chronic conditions and multimorbidity” subsection as it is. In the case that the questions for obtaining the response of participants regarding dyslipidemia, hearth problems, chronic lung diseases, digestive diseases, and others included, were different or included specifications during the participant’s interview, including the corresponding specific questions and specifications would improve the utility of the supplementary table.

If conditions with prevalence below 3% were excluded from the analysis (page 3, paragraph 5, sentence 7), it could be stated as part of the design of the study.

Page 4, paragraph 1, sentence 1. Please, verify type error: “In this study, For ease…”

Please verify table 1 for duplicated “n (%)”, and possibly misplaced between the variables “gait speed” and “age grouped.”

Discussion:

The discussion is speculative. Consider that the analysis described is cross-sectional while re-writing the section.

Organize the discussion as suggested by Docherty M, Smith R. The case for structuring the discussion of scientific papers. BMJ. 1999;318(7193):1224-5.

Include as a limitation the exclusion of a subsample of persons missing data for physical performance (page 2, method section, study population subsection). Elaborate on the implication of such elimination in the representativity of the sample. If possible, compare sociodemographic and health characteristics to explore potential health and cognitive bias.

Similarly, the elimination of those participants unable to hold the handgrip dynamometer and to perform the strength testing with both hands (Page 3, paragraph 2, sentence 6) should be considered and discussed as a limitation.

Author Response

Thanks for your valuable comments. We have provided point-to-point responses to the the comments in this letter (Please see the attachment) and made the corresponding changes, all highlighted in red in the text, to the revised manuscript. 

Reviewer 3 Report

Thank you very much for the opportunity to review this valuable manuscript.

I think the author’s perspective is a very important perspective in practice involving elderly individuals. However, regarding the result or conclusion of the study, there is some suspicion in terms of the novelty. Please re-evaluate your considerations in light of the following.

The association between multi-morbidity and poor physical function is well known in terms of prior research regarding frailty. I interpret the novelty of this study to be that the patterns of multiple diseases have been determined and the relationship between each pattern and physical functions has been compared.

In the conclusion (abstract and text), although the author described that “Grip strength and gait speed could be targeted to routinely measure clinical performance among older adults with multimorbidity in clinical settings,” measurement of physical function is essential in the diagnosis of frailty. Since frailty is included in the concept in terms of disease overlap, the conclusion as it stands does not seem to be novel in this manuscript. I would like to ask you to describe the conclusions that can be deduced from the results of this study with reference to articles on frailty.

Moreover, I interpreted the results of this study to be uniformly dependent on the number of diseases, without any characteristics of the pattern. In the discussion, the author mentioned that “we observed some differences in associations between physical performance and various multimorbidity patterns,” but I think it is necessary to explain the results in detail and the mechanism that you speculate about this point. (I am aware that the causal relationship is unclear.)

Author Response

(The authors gave the same response as above.)

Reviewer 4 Report

Dear authors, it is a pleasure for me to be able to review a work on such an interesting and current topic.

The effort devoted to the realization of the manuscript is appreciated, however, there are some issues I would like to point out:

First, I suggest a modification of the title, starting directly with Associations...

Secondly, the introduction, although well prepared, lacks a deeper understanding of the diseases suffered by adults and the possible associations between them. I would, therefore, suggest extending it accordingly.

I have no comments on the section on materials and methods.

With regard to the section on results, I suggest revising table 1 and dividing it into two tables, in order to make it easier to read and understand.

Finally, as regards the conclusions of the manuscript, I would personally develop it further on the basis of the different associations found in the work. It is important to comment on the effects and the need for public intervention programs, but it is also important to highlight the fundamental findings of the work.

As far as the bibliography is concerned, the authors have to check that the style used is adequate as stipulated by the journal in question. Check all references.

Author Response

(The authors gave the same response as above.)

Round 2

Reviewer 1 Report

No further comments.

Author Response

Thanks for your kind review!

Reviewer 2 Report

Dear authors!
Thank you for addressing the comments of the previous review. I have read the revised version of the manuscript titled “The association between multimorbidity and physical performance in older Chinese adults” finding the revised version improved in clarity. However there are some issues to address.

The values for gait speed are different in the abstract and the methods section (page 1, line 17-18 vs page 4, line 160).

Please verify the expression “…from1.26…” at page 1 line 26.

Please clarify the sentence at page 3, lines 106-108.

Please verify the word “…patter…” at page 5, line 196.

Author Response

Dear reviewer,

Thanks for your kind review and reply. We have corrected the errors we made and sorry for the mistakes. We have made point-to-point response as below:

The values for gait speed are different in the abstract and the methods section (page 1, line 17-18 vs page 4, line 160).

Response: Thanks for your feedback. We have correct the error made in the abstract section.

Please verify the expression “…from1.26…” at page 1 line 26.

Response: Thanks for your feedback. We have correct the error made in the abstract section.

Please clarify the sentence at page 3, lines 106-108.

Response: Thanks for the comment. We have deleted the sentence as we found it was repetition with the sentence “A generalized estimating equation was used to examine the relationships of individual chronic conditions, condition count, and multimorbidity patterns with physical performance including grip strength and gait speed.” in the Statistical analysis section.

Please verify the word “…patter…” at page 5, line 196.

Response: Thanks for your feedback. We have correct the error we made.

Reviewer 3 Report

Thank you for revising the manuscript. I have no corrections to make.

Author Response

Thanks for kind review!